# Cryoprotective Metabolites Are Sourced from Both External Diet and Internal Macromolecular Reserves during Metabolic Reprogramming for Freeze Tolerance in Drosophilid Fly, *Chymomyza costata*

**DOI:** 10.3390/metabo12020163

**Published:** 2022-02-09

**Authors:** Martin Moos, Jaroslava Korbelová, Tomáš Štětina, Stanislav Opekar, Petr Šimek, Robert Grgac, Vladimír Koštál

**Affiliations:** 1Institute of Entomology, Biology Centre, Czech Academy of Sciences, Branišovská 31, 370 05 České Budějovice, Czech Republic; moos@bclab.eu (M.M.); j.korbelova87@centrum.cz (J.K.); tomas.stetina@seznam.cz (T.Š.); opekar@bclab.eu (S.O.); simek@bclab.eu (P.Š.); robert.grgac@entu.cas.cz (R.G.); 2Faculty of Science, University of South Bohemia, 370 05 České Budějovice, Czech Republic

**Keywords:** cryoprotectant metabolites, metabolic pathways, transcriptomics, metabolomics, proline, trehalose, betaine

## Abstract

Many cold-acclimated insects accumulate high concentrations of low molecular weight cryoprotectants (CPs) in order to tolerate low subzero temperatures or internal freezing. The sources from which carbon skeletons for CP biosynthesis are driven, and the metabolic reprogramming linked to cold acclimation, are not sufficiently understood. Here we aim to resolve the metabolism of putative CPs by mapping relative changes in concentration of 56 metabolites and expression of 95 relevant genes as larvae of the drosophilid fly, *Chymomyza costata* transition from a freeze sensitive to a freeze tolerant phenotype during gradual cold acclimation. We found that *C. costata* larvae may directly assimilate amino acids proline and glutamate from diet to acquire at least half of their large proline stocks (up to 55 µg per average 2 mg larva). Metabolic conversion of internal glutamine reserves that build up in early diapause may explain the second half of proline accumulation, while the metabolic conversion of ornithine and the degradation of larval collagens and other proteins might be two additional minor sources. Next, we confirm that glycogen reserves represent the major source of glucose units for trehalose synthesis and accumulation (up to 27 µg per larva), while the diet may serve as an additional source. Finally, we suggest that interconversions of phospholipids may release accumulated glycero-phosphocholine (GPC) and -ethanolamine (GPE). Choline is a source of accumulated methylamines: glycine-betaine and sarcosine. The sum of methylamines together with GPE and GPC represents approximately 2 µg per larva. In conclusion, we found that food ingestion may be an important source of carbon skeletons for direct assimilation of, and/or metabolic conversions to, CPs in a diapausing and cold-acclimated insect. So far, the cold-acclimation- linked accumulation of CPs in insects was considered to be sourced mainly from internal macromolecular reserves.

## 1. Introduction

Convergent evolution repeatedly ‘discovered’ benefits of accumulation of a stereotypic family of cytoprotectant molecules (sugars and polyols, amino acids and derivatives, methylamines) in various organisms from archaea and bacteria to animals responding to different environmental stressors such as heat, cold, freezing, drought, hypersalinity, or high hydrostatic pressure [1,2,3,4]. For instance, the insects that invaded temperate and polar habitats have evolved accumulation of low molecular weight cryoprotectants (CPs) as one of principal tenets of their complex adaptive strategies for survival at subzero temperatures [5,6,7,8]. Some insects avoid internal freezing by extensive supercooling of their body liquids, others tolerate extracellular ice formation, and still others avoid freezing by losing most of their body water [5,9,10,11,12]. Whatever strategy the insect uses, the accumulation of CPs is considered to be one of the major mechanisms that protects insect cells against damage exerted by low temperatures (supercooling), growing ice crystals and freeze-dehydration (freeze tolerance), or evaporative loss of body water (cryoprotective dehydration) [5,6,7,8]. The metabolic origin of insect CPs is relatively well understood for the class of sugars and polyols [13,14,15,16,17,18] but was not sufficiently studied for the classes of amino acids and methylamines. Here we contribute by tracing the sources of carbon skeletons and the pathways for biosynthesis of all three CP classes in the extremely freeze-tolerant drosophilid fly, *Chymomyza costata*. The larvae of this fly survive winter in cold-temperate and sub-arctic climate zones in a state of deep dormancy (diapause), during which they acquire extremely high levels of freeze tolerance. In the laboratory, the diapause cold acclimated larvae can even survive long-term cryopreservation in liquid nitrogen (LN_2_), which makes them an interesting model for cryobiology [19,20,21,22].

Most insects accumulate CPs seasonally. Their metabolic pathways are hormonally diverted from support of active life toward preparation for winter dormancy during the relatively warm end of summer or autumn [23,24,25,26]. In autumn, insects start to accumulate macromolecular substrates such as glycogen, proteins and lipids [27]. Later in the season, usually upon cold acclimation during the coming winter, the macromolecular substrates may serve as sources for CP biosynthesis [28,29]. This general mechanism is best exemplified by degrading glycogen reserves in order to produce various sugars and polyols such as glucose, trehalose, glycerol, sorbitol and ribitol in different cold-hardy animals, including insects [13,14,15,16,17,18]. Primary control over this process is at the level of glycogen breakdown via direct triggering of glycogen phosphorylase activity by a drop of temperature below 5 °C [28,30]. Cold-activation of glycogen phosphorylase goes hand in hand with concerted changes in expression and allosteric modulation of activity in many other enzymes of intermediary metabolism, which finally results in accumulation of sugars and polyols [31,32,33,34]. Similarly to animals, cold-hardy plants also accumulate soluble sugars [35] by degrading their macromolecular reserves of starch [36].

Some insects seasonally accumulate high concentrations of free amino acids, especially proline [37,38,39,40]. The cryoprotective effects of proline are well documented [19,20,41,42,43,44,45] but the metabolic origin of proline in cold-hardy insects was not previously studied. We speculated that collagens in the extracellular matrix (ECM) may represent macromolecular depots of relatively ‘dispensable’ proteins from which proline, and other amino acids, can be obtained upon need [20,40]. Collagens are known to make up a large part of total proteins in animals, and are characterized by a high content of proline and hydroxyproline in their structure [46,47]. For example, massive degradation of ECM collagens is known to supply proline as an alternative energy source for rapid metabolism of proliferating or cancerous animal cells [48]. Proline appears to be one of the most widely distributed cytoprotectants accumulated under various stress conditions not only in insects and marine invertebrates [3] but also in eubacteria, protozoa, algae and plants [49,50]. The strong evidence for the role of proline in plant freeze tolerance comes from genetic manipulations of *Arabidopsis* [51]. Plants and eubacteria synthesize proline from glutamate and ornithine precursors in response to stress [49]. 

In addition, we found recently that cold-acclimated larvae of *C. costata* accumulate glycerophosphocholine (GPC), glycerophosphoethanolamine (GPE) and methylamines such as glycine betaine (betaine henceforth), and sarcosine [22]. Accumulation of betaine is well documented in plants and halophilic prokaryotes under various types of environmental stress [52,53], and genetic proofs for betaine contributing to freeze tolerance exist [54,55]. Betaine and GPC also provide osmoprotection of mammalian kidney cells exposed to high salt concentrations [56,57]. Animals, plants, and bacteria synthesize betaine from choline via a two-step oxidation pathway [56,58] and may further convert betaine to DMG (dimethylglycine) and sarcosine. Some halophilic prokaryotes, however, can synthesize betaine from glycine via a three-step methylation, which requires activity of two specific methyltransferases [59].

Here, we explore adaptive adjustments in metabolism as the larvae of *C. costata* transit from freeze-sensitive to freeze-tolerant phenotype during entry into winter diapause and subsequent cold acclimation. We map onto metabolic pathways the changes in relative quantities of 56 metabolites (analyzed using liquid chromatography and high resolution mass spectrometry, LC-HRMS), and relative expression of 95 key genes coding key metabolic enzymes and transport systems (analyzed using reverse transcription quantitative PCR, RT-qPCR). We cover metabolic pathways representing all three major chemical groups of putative CPs. We focus on distinguishing between alternative sources of carbon skeletons for the CPs’ biosynthesis, as follows. (i) Proline (and also other amino acids): is it derived from stored macromolecular reserves such as collagens and total proteins or from metabolic precursors such as glutamine, glutamate and ornithine, or rather assimilated from diet as an external source? (ii) Trehalose (and other sugars and polyols): derived from glycogen reserves or dietary starch and glucose? (iii) Methylamines: choline coming from interconversions of phospholipids or glycine which can be massively released during the breakdown of collagens?

## 2. Results and Discussion

We conducted experiments using four acclimation/phenotypic variants of 3rd instar *C. costata* larvae that were generated according to previously published acclimation protocols [19,20,21] and widely differed in freeze tolerance and cryopreservability in LN_2_. All acclimation variants are described in detail in the Materials and Methods and presented in Table 1. Briefly, the variants SD3, SD6, and SD11 represent successive stages of diapause maintenance (3, 6, and 11 weeks old larvae, respectively) under constant warm conditions (18 °C). During diapause maintenance, relatively weak freeze tolerance gradually develops. The variant SDA represents the 11 weeks old larvae that undergo a gradual cold-acclimation process (11 °C for 1 week, followed by 4 °C for 4 weeks), which results in development of a maximal level of freeze tolerance.

We restricted analyses to changes occurring *within* diapause larval phenotypes, focusing on the transition from freeze sensitivity (SD3) to extreme freeze tolerance including the ability to survive long-term cryopreservation in LN_2_ (SDA). We omit metabolic reconstructions for non-diapause larvae as they exhibit intense locomotive activity, voracious feeding, and rapid growth and development; a stark contrast to diapausing larvae which are characterized by metabolic suppression and hormonally arrested development [26,60]. Such deep phenotypic differences between non-diapause and diapause could overwhelm or obscure the metabolic changes associated with cold acclimation and the acquisition of freeze tolerance. We mapped changes in 56 metabolites (Appendix A) and 95 genes (Appendix A) onto three schematic pathways (proline, trehalose, betaine) and will discuss them one by one. It is important to note that our schematic maps do not represent any specific cell type though they show plasmatic membranes and distinguish between cytosolic and extracellular spaces. As the metabolomic analyses were conducted using whole larvae, resolution to tissue level is not possible. The metabolomics results presented in this study are based on relative quantification (comparing chromatographic peak areas). We will refer to our earlier studies [20,22] for absolute quantities of select metabolites.

### 2.1. Proline and Other Amino Acids

Complete results of metabolomics and transcript analysis are presented in Appendix A while the differences between diapause maintenance (transition from SD6 to SD11) and cold acclimation-linked metabolisms (transition from SD6 to SDA) are graphically presented in Figure 1.

Larvae undergoing cold acclimation exhibited more intense up- or down-regulations of proline-associated metabolites and transcripts compared to those in diapause maintenance (compare Figure 1a vs. Figure 1b). The chromatographic peak area of proline increased 1.6-fold during cold acclimation, and we observed concomitant peak area increases for glycine (2.2-fold) and hydroxyproline (2.1-fold) (Figure 1b, Appendix A). The synchronous accumulation of glycine, proline, and hydroxyproline suggests that degradation of collagens—which are characterized by a high content of just these three amino acids [46]—is a potential source of accumulated proline. For example, the *C. costata* collagen IVα1 (Seq33535, Appendix A) contains approximately twice as many glycine residues as proline residues (it has 1782 aa-residues and a molecular mass of 174,993 kDa; 497 residues (27.9%) are glycines; 244 residues (13.7%) are prolines and hydroxyprolines in an unknown ratio). Hydroxyproline is synthesized by proline hydroxylases (genes no. 1 and 2 in Figure 1) as a post-translational modification of proline residues bound to nascent collagen chains [61], so it is unlikely that the accumulated hydroxyproline originated from sources other than collagen-like proteins. However, other pieces of evidence speak clearly against the probability that collagens are the major source of accumulated proline, which we discuss in detail below.

First, the absolute concentration of glycine in SDA larvae is at least two orders lower than the concentration of proline [20,22]. Degradation of collagens, however, should release approximately twice as many glycine residues as proline residues. Should the proline be released from collagens, the massive flow of concomitantly released glycine would have to be catabolized or, potentially, converted to betaine (we discuss this possibility later). Second, a stoichiometric calculation shows that *C. costata* larvae do not carry stores of degradable protein sufficient to account for the amount of free proline accumulated by the SDA phenotype. We may consider that: (i) the average SDA-larva accumulates proline at a concentration of 339 mmol.kg^−1^ of body water while an average SD11-larva has only 162 mmol.kg^−1^ of proline [20], which corresponds to a total pool of 475 nmoles (54.6 µg) or 227 nmoles (26.1 µg) of proline per larva, respectively (considering larval FM as 2 mg and water content as 1.4 mg); (ii) in order to produce 475 − 227 = 248 nmoles of proline, the average larva would need to degrade at least 178 µg of collagen IVα1 (considering that degradation of 1 mole, i.e., almost 175 kg, of collagen IVα1 produces 244 moles of proline and hydroxyproline residues); (iii) the average SD6-larva, however, carries only ca. 270 µg of total protein (450 µg mg^−1^ DM), and only 38 µg of total protein is depleted during cold acclimation (Figure 2a). Third, the overall content of collagens increased during cold acclimation (Figure 2b), while a decrease would be expected if proline was released by degradation of collagens. Paradoxically, the activity of collagen-degradation enzymes (matrix metalloproteinases; MMPs) increased significantly (albeit slightly) during cold acclimation as well (Figure 2c). Fourth, using analysis of total proteins on PAGE, we verified that dominant protein bands show stable patterns across phenotypic variants suggesting that proteins are not massively degraded (Appendix A). Fifth, transcriptomic analysis suggested that proline transport is bolstered in SDA larvae as we observed significant (1.9-fold) increase in the relative expression of the gene coding for amino acid nutrient transporter NAAT1 (no. 11, Figure 1b, Appendix A) which shows high specificity for proline [62]. Nevertheless, the origin of proline from collagen was not supported by transcriptomics as there were: (i) 3.3-fold decrease in relative expression of the gene coding for collagen IVα1 (no. 3, Figure 1b, Appendix A); (ii) 1.8-fold decrease in one of two genes coding for MMPs (*MMP2*, no. 6, Figure 1b, Appendix A); countered by (iii) 1.7-fold increase in Timp (no. 4, Figure 1b, Appendix A) which is coding for an inhibitor of MMPs [63].

Similarly as in plants under stress [49], proline in *C. costata* may be sourced from its nearest metabolic precursors, glutamine, glutamate and ornithine. We found that relatively large reserves of glutamine build up in early diapause (transition from SD3 to SD6), while they are partially depleted during subsequent cold acclimation (Figure 1b, Appendix A). In absolute terms, total glutamine content may reach up to 180 nmoles in the average SD11-larva while only 38 nmoles are found in the average SDA-larva [20]. Hence, the conversion of 180 − 38 = 142 nmoles of glutamine may help to explain the origin of almost one half of the proline molecules accumulated in SDA-larvae, while the second half remains unexplained. The activation of glutamine conversion via the glutamate pathway is supported by upregulated glutamate synthetase (no. 30) and P5C reductase (no. 24) expressions during cold acclimation (Figure 1b). The glutamate synthetase activity, however, requires a supply of additional carbon skeletons from ketoglutarate, which might be limited due to general suppression of glycolysis and TCA metabolism in diapausing, cold-acclimated larvae (see discussion on trehalose metabolism). Stimulation of ornithine conversion via upregulated expression of ornithine aminotransferase (no. 35) may also contribute to accumulation of proline (Figure 1b). However, the total pools of glutamate (ca. 8 nmoles) and ornithine (ca. 0.5 nmoles) are relatively low throughout the acclimation variants [20], which suggests that these compounds are metabolic intermediates rather than sources from which the accumulation of proline would drive the carbon skeletons.

Since internal sources (either proteins or metabolic precursors) are obviously not sufficient to fully explain a massive accumulation of proline, the larval diet should be another source. Previously, we had not considered the diet as a source of proline because diapausing insects are known to drastically suppress their feeding, digestion, and metabolic rates [27,64]. The larvae of *C. costata*, for instance, gradually lose weight during diapause [65]. However, examining the feeding activity of diapausing *C. costata* larvae, we found that they consume food not only during diapause maintenance at a relatively high temperature of 18 °C but also during cold acclimation to 4 °C (Figure 3). 

Next, we analyzed the composition of the larval diet (Appendix A) and found that proline was the highest in concentration of all the amino acids (2.7 mmol.kg^−1^ FM or 25.6 mmol.kg^−1^ DM). The amount of dietary proline that is directly ‘assimilated’ from food depends on the amount of food ingested and the food assimilation efficiency, both unknown for *C. costata* larvae. At 100% assimilation efficiency, the average larva of *C. costata* would need to ingest 37 mg of diet FM in order to accumulate 100 nmoles of proline. At a more realistic assimilation efficiency of 16% as demonstrated for larvae of the black soldier fly *Hermetia illucens* [66] this amount would increase to 231 mg of diet FM. Because the fresh mass of *C. costata* larvae gradually decreases during diapause at low temperatures [65], the calculations suggest that larvae must be able to absorb proline from the ingested diet without assimilating many of the other nutrients. The high ability of *C. costata* larvae to ‘harvest’ dietary proline is supported by two earlier observations: (i) the non-diapause larvae were able to increase their whole body concentration of proline almost 7-fold (from 39 to 268 mmol.kg^−1^ FM) within just three days of feeding on proline augmented diet [22]; (ii) when feeding *Drosophila melanogaster* larvae the same proline-augmented diet as *C. costata*, we found that *D. melanogaster* accumulated about one order less proline relative to *C. costata* [41]. In addition, glutamate is the second most concentrated amino acid in larval food (2.6 mmol.kg^−1^ FM or 24.9 mmol.kg^−1^ DM) (Appendix A) and, hence, may serve as additional dietary precursor for biosynthesis of proline.

Overall, our results suggest that high concentrations of proline in SDA larvae of *C. costata* originate from different sources: (i) direct assimilation of ingested dietary proline, together with metabolic conversion of dietary glutamate (a proline precursor), can explain about one half of the proline accumulation; (ii) metabolic conversion of larval glutamine reserves formed in early diapause can explain the second half of proline accumulation; (iii) two other minor sources can be the metabolic conversion of ornithine and the degradation of larval ECM collagens and other proteins. 

### 2.2. Trehalose and Other Sugars and Polyols

Complete results of metabolomics and transcript analysis are presented in Appendix A, while the differences between diapause maintenance and cold-acclimation-linked metabolisms are graphically presented in Figure 4. Trehalose consists of two glucose molecules [67] and its non-reducing nature allows for safe ‘storage’ in large amounts in larval hemolymph. No trehalose accumulation was observed in *C. costata* during diapause maintenance (Figure 4a, Appendix A) but the chromatographic peak area of trehalose increased 2.2-fold during cold acclimation (Figure 4b, Appendix A). In support of upregulated trehalose biosynthesis in SDA larvae, the expression of a gene coding for two synthetic enzymes, *trehalose 6-phosphate synthase* (gene no. 20) and *phosphatase* (no. 21) increased substantially (4.1- and 10.1-fold, respectively), while the expression of a gene coding for the catabolic enzyme *trehalase* (no. 23) decreased 4.1-fold during cold acclimation (Appendix A). 

The metabolic map of trehalose further suggests that the glycolytic flux toward TCA was suppressed during late diapause maintenance and especially during cold acclimation (Figure 4, Appendix A). In SDA-larvae, the genes coding for glycolytic enzymes nos. 5, 6, 7, and 8 were downregulated as were intermediates of the fermentation and TCA cycle: pyruvate, lactate, citrate, ketoglutarate, and fumarate. In addition, ATP concentrations increased whereas the concentrations of ADP and AMP decreased (Figure 4b), which is consistent with a relatively low demand for energy turnover during diapause and even less at low temperatures [27,64]. Glucose metabolism was redirected from glycolysis to: (i) trehalose biosynthesis (discussed above); (ii) sorbitol biosynthesis (no. 9 aldoketoreductase was upregulated 4.2-fold, resulting in a 2.1-fold increase in sorbitol); (iii) conversion to gluconolactone (no. 11 glucose 6-phosphate dehydrogenase was upregulated 3.5-fold, resulting in a 5.1-fold higher accumulation of 6-P gluconate). The conversion in (iii) is associated with the production of reducing equivalents in the form of NADPH. This reducing power is required, for example, for syntheses of proline from P5C and sorbitol from glucose.

The map of trehalose metabolism shows that glucose was also accumulated during cold acclimation (Figure 4b). Note, however, that glucose reached absolute molar concentrations two orders lower than trehalose [20]. Glycogen might be an important source of glucose units as suggested by a trend of total glycogen accumulation during early diapause, followed by its partial depletion during cold acclimation (Figure 5a). A similar depletion trend has been stereotypically observed in many diapausing insects, and glycogen is thought to be the major source for accumulation of various sugar-derived CPs in insects [13,28,30]. The total pool of trehalose is approximately 79 nmoles (27 µg) in an average SDA-larva, while only 34 nmoles 9 (11.6 µg) are found in an SD11-larva [20]. The accumulation of 79 − 34 = 45 nmoles of trehalose can be explained if only breakdown of glycogen is considered. We may assume that: (i) the cold-acclimation-linked depletion of glycogen represents 34.3 µg.mg^−1^ DM (Figure 5a i.e., 17.1 µg per larva); (ii) this amount of glycogen can release about 145 nmoles of glucose, from which about 72 nmoles of trehalose can be synthesized. The gene coding for a key enzyme of glycogen degradation, *glycogen phosphorylase* (no. 17) showed little change in expression upon cold acclimation (a 1.3-fold decrease, Appendix A) and the total enzymatic activity of glycogen phosphorylase was significantly lower in SDA-larvae compared to other phenotypes. At least, we observed a relatively high contribution of the active ‘a’ form to total glycogen phosphorylase activity in cold-acclimated SDA larvae (Figure 5b). This is in accordance with the well-described process of cold activation of glycogen phosphorylase that initiates synthesis of sugar-derived CPs in insects [32]. 

In addition to the internal source of glucose units (glycogen), large amounts of glucose are available in the larval diet (11.6 mmol.kg^−1^ FM or 109.9 mmol.kg^−1^ DM; Appendix A). Thus, the ingested diet may serve as an additional source of glucose for diapausing, cold-acclimated larvae. The diet also contains starch and maltose (maltose in concentration of 3.6 mmol.kg^−1^ FM or 33.9 mmol.kg^−1^ DM; Appendix A), and the transcriptomic data suggest that digestion of starch and maltose might be enhanced in SDA larvae (see the increasing expression of amylases and maltase, nos. 1 and 3 in Figure 4b). However, this was not confirmed by direct assay of enzyme activities; the combined activity of general amylases and maltases decreased sharply during cold acclimation (Figure 5c). Overall, our results suggest that glycogen is the main source of glucose units for trehalose synthesis and accumulation, whereas the diet may serve as an additional source.

### 2.3. Betaine, Other Methylamines and Related Compounds

Complete results of metabolomics and transcript analysis are presented in Appendix A, while the differences between diapause maintenance and cold acclimation-linked metabolisms are graphically presented in Figure 6. During cold acclimation, the chromatographic peak area of betaine increased 1.6-fold and the gene coding for betaine transporter (no. 12) was upregulated 2-fold (Figure 6b, Appendix A). In contrast, betaine decreased slightly (by 1.3-fold) and no change in transporter expression was observed during diapause maintenance (Figure 6a, Appendix A). The absolute concentration of betaine in hemolymph of SDA-larvae (6 mmol.L^−1^, [22] ) was relatively low compared to that observed in plants under various types of environmental stress: tens to hundreds of mmol.kg^−1^ [52]; or halophilic prokaryotes: over 1 mol.kg^−1^ [53]. Nevertheless, genetic manipulations of *Arabidopsis* metabolism show that accumulations of just millimolar concentrations of betaine significantly improve plant tolerance to various stresses including cold and freezing [54,55]. 

Animals, plants, and bacteria synthesize betaine from choline via a two-step oxidation pathway [56,58] and may further convert betaine to DMG (dimethylglycine) and sarcosine (methylglycine) by activities of BHMT and DMGDH enzymes (see Figure 6 for more details). Only some halophilic prokaryotes can synthesize betaine from glycine via a three-step methylation, which requires specific activity of two methyltransferases (EC 2.1.1.156 and EC 2.1.1.161) [57]. Because the potential degradation of collagens would produce a large excess of glycine (discussed above) and because cold acclimation is accompanied by concerted increases in sarcosine (methylglycine), dimethylglycine, and betaine (trimethylglycine) concentrations (Figure 6b), we investigated whether a ‘halophile-like’ three-step methylation pathway contributes to betaine biosynthesis in *C. costata* larvae. Using fluxomics LC-HRMS analysis of ^13^C-labelled compounds, we found no support for a halophile pathway in *C. costata* i.e., conversion of ^13^C-glycine beyond sarcosine was not possible in whole non-diapause (LD) larvae (Figure 7) or in isolated fat body and muscle dissected from non-diapause and diapause (SDA) larvae (Appendix A). In contrast, we found good support for the classical animal pathway; i.e., ^13^C-choline was converted not only to betaine but also to DMG, sarcosine, and glycine (Figure 7 and Appendix A).

The origin of choline itself, however, remains unclear. It was shown that insects cannot synthesize choline and this must be supplied by the diet [68,69]. Nevertheless, the presence of an ortholog of *GPC phosphodiesterase* gene in genomes of *D. melanogaster* (CG2818) or *C. costata* (no. 9 in Figure 6, Appendix A) suggests that flies can release choline from GPC, similarly to its release in mammals [70].

The consistently increasing concentrations of GPC and GPE during diapause maintenance and cold acclimation (Appendix A) are likely related to the remodeling of membrane phospholipids that accompanies the entry into diapause and cold acclimation of *C. costata* larvae [71]. The increases in GPC an d GPE concentrations were consistent during transition from SD3 to SDA but often <0.58 log_2_-fold (Appendix A) during each step of acclimation, which is why they do not appear in red symbols in the metabolic map (Figure 6b). GPC together with sorbitol, betaine, and myo-inositol are believed to play a role as compatible osmolytes in kidney medulla cells, counteracting the high interstitial concentrations of perturbing Na^+^ ions and urea [70,72,73]. The GPC concentrations in the inner kidney medulla of various mammals reach tens of mmol.kg^−1^ FM [74]. The absolute concentrations of GPC (2.9 mmol.L^−1^) and GPE (4.0 mmol.L^−1^) in SDA-larval hemolymph [22] are much lower than in mammalian kidney. 

We have not measured the absolute concentrations of GPC, GPE and methylamines in larval whole body in our earlier work [20]. Based on hemolymph and tissue concentrations, however, we can estimate sum mass of accumulated GPC, GPE, betaine, and sarcosine as representing approximately 10 moles per SDA larva (corresponding to approximately 2 µg). Though these absolute amounts are relatively low compared to proline or trehalose, the role of GPC, GPE and methylamines in cryoprotective mixtures deserves further investigation in future. Collectively, our results suggest that GPC and GPE are metabolized and accumulated in *C. costata* larvae in association with membrane phospholipid conversions. Betaine is synthesized, as in other animals, from choline via a two-step oxidation pathway.

## 3. Materials and Methods

### 3.1. Insects

A colony of *C. costata*, Sapporo strain (Riihimaa and Kimura, 1988) was reared on an artificial diet in MIR 154 incubators (Sanyo Electric, Osaka, Japan) as described previously [75,76]. Four phenotypic variants (SD3, SD6, SD11, and SDA, Table 1) of the 3rd instar larvae were generated according to our earlier acclimation protocols [19,20] as described in Figure 8. 

### 3.2. Extraction of Metabolites and LC-HRMS Platform

Analyses were performed for whole *C. costata* larvae (pools of 5 larvae taken in 4 replicates). The larvae were weighed to obtain fresh mass, plunged into LN_2_, and stored at −80 °C until analysis. Frozen samples were melted on ice and homogenized in 400 μL of extraction buffer: methanol:acetonitrile:deionized water mixture (2:2:1, *v*/*v*/*v*). The methanol and acetonitrile (Optima™ LC/MS) were purchased from Fisher Scientific (Pardubice, Czech Republic) and the deionized water was prepared using Direct Q 3UV (Merck, Prague, Czech Republic). Internal standards, *p*-fluoro-DL-phenylalanine, methyl α-D-glucopyranoside (both from Sigma-Aldrich, Saint Luis, MI, USA) were added to the extraction buffer, both at a final concentration of 200 nmol.mL^−^^1^. Samples were homogenized using a TissueLyser LT (Qiagen, Hilden, Germany) set to 50 Hz for 5 min (with a rotor pre-chilled to −20 °C). Homogenization and centrifugation (at 20,000× *g* for 5 min at 4 °C) was repeated twice and the two supernatants were combined. 

In the whole-body extracts, we performed relative quantification analyses of 56 select metabolites (listed in Appendix A) using the LC-HRMS platform on the Q Exactive Plus high resolution Orbitrap mass spectrometer coupled to a Dionex Ultimate 3000 liquid chromatograph and a Dionex open autosampler (all from ThermoFisher Scientific, Waltham, MA, USA). Full scan LC-HRMS positive and negative ion mass spectra were recorded in separate runs with a mass range of 70–1000 Da at 70,000 resolution (at mass *m*/*z* 200). The LC-HRMS settings were: scan rate at ±3 Hz, 3 × 106 automatic gain control (AGC) target, and maximum ion injection time (IT) 100 ms. Source ionization parameters were as follows: (±) 3000 kV spray voltage, 350 °C capillary temperature, sheath gas at 60 au, aux gas at 20 au, spare gas at 1 au, probe temperature 350 °C, and S-Lens level at 60 au. For accurate mass identification, we used lock masses of 622.0290 Da for the positive ion mode and 301.9981 Da for the negative ion mode. Chromatographic separation of metabolites was carried out on the SeQuant ZIC-pHILIC (150 mm × 4.6 mm i.d., 5 μm, Merck, Darmstadt, Germany), the mobile phase flow rate was 450 μL/min; the injection volume, 5 μL; column temperature, 35 °C. The mobile phase: A = acetonitrile (ThermoFisher Scientific, Waltham, MA, USA). B = 20 mM aqueous ammonium carbonate (pH = 9.2 adjusted by NH_4_OH, Sigma-Aldrich); gradient: 0 min, 20% B; 20 min, 80% B; 20.1 min, 95% B; 23.3 min, 95% B; 23.4 min, 20% B; 30.0 min 20% B. Data were acquired and metabolites identified using an in-house Metabolite Mapper platform equipped with an internal metabolite database in conjunction with Xcalibur™ software (v2.1, ThermoFisher Scientific, Waltham, MA, USA). All 56 metabolites were quantified relatively using the areas under respective chromatographic peaks normalized to fresh mass of larval samples. 

### 3.3. Target Gene Sequences and Gene Expression Analysis 

Sequences putatively coding for 95 target genes (listed in Appendix A) involved in CP metabolism, and five reference genes, were retrieved from a *C. costata* Illumina RNAseq database (ArrayExpress accession E-MTAB-3620) published earlier [77]. Using the Geneious R9.1.8 platform (Biomatters Ltd., Auckland, New Zealand), we used the BLASTn tool to align the *C. costata* target gene sequences with those most similar to sequences of *D. melanogaster* (Appendix A). Next, we designed optimal oligonucleotide PCR primer pairs for all sequences using Geneious, observing uniform criteria such as small PCR product size (70–200 bp), optimal size of primer (20 bp), optimal primer T_m_ (61 °C), optimal percentage of GC (50%), etc. (all primer pairs given Appendix A). 

We used RT-qPCR to compare the relative gene expressions of four diapause larval phenotypes (SD3, SD6, SD11, SDA) as described earlier [77]. LD larvae were excluded from these comparisons, as substantial differences in their gene expression relative to SD larvae [26] would obscure our ability to detect differences among SD variants. We extracted total RNA from larvae (pools of 30 larvae were taken in 3 biological replicates) using RNA Blue (Top-Bio, Vestec, Czech Republic), levelled to exactly 1 µg.µL^−1^ sterile water and treated with DNase I (Invitrogen, ThermoFisher Sci., Prague, Czech Republic). We then converted 5 µg of total RNA to cDNA using Superscript III (Invitrogen, Carlsbad, CA, USA). The cDNA products (20 μL) were diluted 25 times with sterile water. We performed qPCR with a CFX Connect PCR Cycler (BioRad, Philadelphia, PA, USA), using the oligonucleotide primers shown in Appendix A [21]. Relative ratios of the target gene mRNA to the geometric mean of the levels of five reference gene mRNAs were calculated as dd*C*_T_ [26,78].

### 3.4. Supplementary Assays to Clarify the Metabolism of CPs

#### 3.4.1. A Brilliant Blue Larval Food Intake Assay 

A Brilliant Blue Larval Food Intake Assay [79] was used to assess whether larvae in diapause ingest food. In a preliminary assay, we verified that 100% of non-diapause (LD) larvae (*n* = 25) will intake larval diet mixed with 0.1% Brilliant Blue R dye (Sigma) (BB diet). The LD larvae were transferred to BB diet for 1 h at constant 18 °C, after which they were observed and photographed under a dissecting microscope. Food intake was apparent as blue coloration of the gut. Next, the larvae of different diapause acclimation variants (SD6, SDA7, and SDA9) were used for the BB assay as shown in Figure 3a.

#### 3.4.2. Total Proteins

Whole *C. costata* larvae (pools of 5 larvae taken in 3 replicates) were homogenized and total proteins were extracted twice in 400 µL of 50 mM Tris buffer, pH 7.2, containing 100 mM NaCl, 1% deoxycholate, 0.2% SDS, and 1% Nonidet. Total proteins were determined by bicinchoninic acid assay (BCA) [80]. 

#### 3.4.3. PAGE

The extracts of total proteins (see above) were levelled to equal protein concentration of 1 mg.mL^−1^, mixed with 4× Loading buffer (LB) in a volume ratio 1:3, and heated at 95 °C for 10 min (the LB contained: 2 mL 1M Tris-HCl, pH 6.8; 4 mL 1M dithiothreitol; 0.8 g sodium dodecylsulphate (SDS); 40 mg bromphenol blue (BPB); and 3.2 mL glycerol in 10 mL of distilled water). Aliquots were loaded (20 µL per lane) onto precast 12% MiniProtean TGX gels (BioRad, Hercules, CA, USA) and separated by polyacrylamide gel electrophoresis (PAGE) using Mini-Protean Tetra Cell apparatus (Biorad). The gels were run in a classical Tris-Glycine-SDS buffer, pH 8.3 at a constant voltage of 150 V until BPB left the gel. Protein bands were stained with Coommassie Brilliant Blue and band densities were measured using Gel Analyzer function of ImageJ software (https://imagej.nih.gov/ij/; accessed on 15 November 2021) (for more information, see Appendix A). 

#### 3.4.4. Total Collagen 

Total collagen in larvae was estimated based on the release of 4-hydroxyproline upon acidic hydrolysis of total larval proteins according to [81]. Whole *C. costata* larvae (pools of 5 larvae taken in 3 replicates) were homogenized in 100 µL of distilled water. Next, 100 µL of concentrated HCl (12 M) was added to the homogenate in a glass tube caped tightly with a Teflon insert. The sample was hydrolyzed at 120 °C for 3 h, after which 5 mg of activated charcoal was added and the sample was centrifuged at 3000 g for 5 min. Then, 50 µL of supernatant was moved to a clean vial and dried at 60 °C under stream nitrogen and stored at −80 °C until LC-HRMS analysis of 4-hydroxyproline was performed.

#### 3.4.5. Collagenase 

Collagenase activity was assayed using the fluorimetric MMP Activity Assay Kit (ab112147, Abcam, Cambridge, UK) according to the manufacturer’s manual. Whole *C. costata* larvae (pools of 5 larvae taken in 3 replicates) were homogenized and total proteins were extracted in 500 µL of 100 mM Tris buffer, pH 7.2. A 25 µL aliquot of supernatant was used as a source of enzymatic activity. We measured the increase in fluorescence over time (15 min at constant 25 °C) at 540/590 nm (Ex/Em) using the microplate reader Infinite 200Pro (Tecan) and normalized the slopes of linear regressions to mg of total protein (BCA assay, see above) in the sample aliquot.

#### 3.4.6. Glycogen

Whole *C. costata* larvae (pools of 5 larvae taken in 3 replicates) were homogenized twice in 400 µL of a methanol:acetonitrile:water mixture. After centrifugation at 20,000× *g* for 10 min at 4 °C, glycogen was extracted from the pellet in hot alkali [82] and quantified by colorimetric assay with phenol and concentrated sulphuric acid [83]. 

#### 3.4.7. Glycogen Phosphorylase

Whole *C. costata* larvae (pools of 10 larvae taken in 3 replicates) were homogenized in 600 µL of 100 mM Tris-HCl buffer, pH 8.0 containing 15 mM mercaptoethanol and 1 mM EDTA. The activity of active (‘a’) and inactive forms (‘b’) of the enzyme were measured in 70 μL aliquots of supernatant as described by [84] and normalized to mg of total protein (BCA assay, see above) in the sample aliquot. The reaction mixture contained 50 mM potassium phosphate buffer, pH 6.8; 5 mg mL^−1^ glycogen (omitted from control), 5 μM glucose-1,6-diphosphate, 0.6 mM NADP, 2 mM 5′AMP, 15 mM MgCl_2_, 2 U mL^−1^ phosphoglucomutase, and 2 U mL^−1^ glucose 6-P dehydrogenase. The active form of the enzyme was measured in the absence of 5′AMP. 

#### 3.4.8. Amylases and Maltases

Whole *C. costata* larvae (pools of 10 larvae taken in 3 replicates) were homogenized in 1 mL of Assay Buffer supplied in the Amylase Assay Kit (ab102523, Abcam, Cambridge, UK) and 50 µL aliquots of supernatant were used as sources of enzymatic activity. We measured the increase of absorbance over time (15 min at constant 25 °C) at 405 nm using the microplate reader Infinite 200Pro (Tecan, Männedorf, Switzerland), converted this to nmoles of released nitrophenol (using the calibration nitrophenol standards), and normalized the results to mg of total protein (BCA assay, see above) in the sample aliquot.

#### 3.4.9. Fluxomics of ^13^C-Labelled Metabolic Precursors of Betaine

We performed assays for (i) whole non-diapause (LD) larvae fed a standard diet augmented with ^13^C-labelled metabolic precursors and (ii) tissues dissected from non-diapause (LD) and diapause (SDA) larvae. The examination of the tissues allowed us to rule out possible contribution of microbes to metabolic conversions in the larval diet or inside the larval gut. The tissues were incubated in Schneider’s *Drosophila* medium (Biosera, Nuaille, France) augmented with ^13^C-labelled metabolic precursors purchased from Cambridge Isotope Laboratories (Tewksbury, MA, USA). The ^13^C-glycine had both carbons labelled (item no. CLM-1017, m.w. 77.05) and the ^13^C-choline chloride had two carbons labelled (1,2-^13^C2 choline chloride, item no. CLM-548, m.w. 141.61).

(i) Whole larvae: suitable doses of ^13^C-labelled precursors in larval diets (^13^C-glycine, 15 mg per g of diet; ^13^C-choline chloride, 20 mg per g of diet) were derived from preliminary experiments showing that lower doses have no effect on the timing of larval development, a ‘suitable’ dose causes a small (2–3 d) delay in timing of pupariation but no mortality, and higher doses cause longer developmental delays and mortality. Third instar LD larvae (15–16 d old) were transferred from a standard diet to a ^13^C-precursor-augmented diet for 6, 24, or 48 h at constant 18 °C (20 larvae per vial with 1 g of diet). Next, larvae were processed for LC-HRMS analysis. 

(ii) Dissected tissues: larval fat body and muscle tissues were dissected from a group of 50 larvae (either LD- or SDA-larvae) into the wells of a 12-well tissue culture plate (TRP, Switzerland) containing 650 µL of Schneider’s medium supplemented with 100 mM of either ^13^C-glycine or ^13^C-choline chloride. After 3 h (LD) or 6 h (SDA) incubation at 22 °C, the tissues were rinsed in Schneider’s medium and processed for LC-HRMS analysis. Results of LC-HRMS analyses were expressed for each target metabolite as a percentage of the peak area of ^13^C-form (subtracting the native occurrence of ^13^C-carbon) relative to the ^12^C-form.

## 4. Conclusions

Here we show that diapausing and cold-acclimated larvae of *C. costata* continue with food ingestion and the diet serves as important source of direct assimilation of, and/or metabolic conversions to, cryoprotective molecules. We show that even the complete degradation of larval reserves of total protein, including collagens, would not suffice to explain accumulation of proline. Moreover, total protein or collagen reserves are depleted only moderately or not at all, respectively, during cold acclimation. We identified direct assimilation of dietary amino acids proline and glutamate, plus metabolic conversion of glutamine reserves, as two major sources of proline accumulation. In agreement with general consensus on metabolic origin of sugar-based CPs, we show that trehalose is sourced mainly from internal glycogen depots that are filled during early diapause and partially depleted during subsequent cold acclimation. Lastly, we suggest that the accumulations of GPC and GPE result from cold-acclimation-linked remodeling of phospholipids. Choline, either released from GPC or assimilated from diet, serves as a source for accumulation of methylamines betaine and sarcosine.

## Figures and Tables

**Figure 1 metabolites-12-00163-f001:**
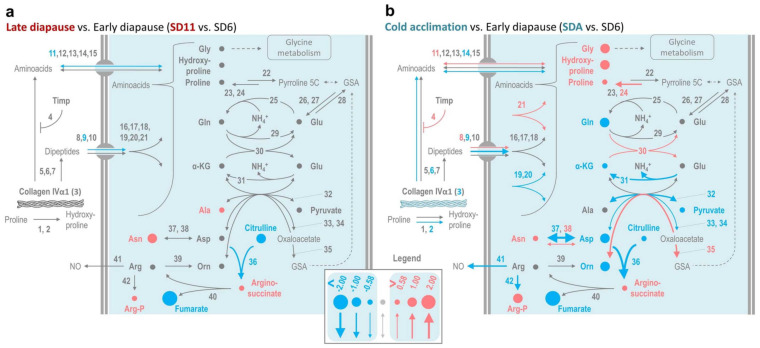
Changes in metabolism of proline during diapause maintenance vs. cold acclimation in *C. costata* larvae. The maps show log_2_-fold changes in relative concentrations of metabolites (circles) and gene expression (arrows) according to numerical results (Appendix A) during diapause maintenance (**a**) and gradual cold acclimation (**b**). Three different thresholds for change were reflected in the size of symbols used (circles and arrows): absolute log_2_-fold change >0.58 (1.5-fold change); >1.00 (2-fold change); and >2.00 (4-fold change). Red indicates upregulation, blue indicates downregulation, grey indicates log_2_-fold changes between −0.58 to 0.58. Genes are coded by numbers (see Appendix A). Thick grey double lines symbolize the plasma membrane with embedded transport systems. α-KG, ketoglutarate; GSA (glutamate-5-semialdehyde) and pyrroline 5C are tautomers and their interconversions are spontaneous.

**Figure 2 metabolites-12-00163-f002:**
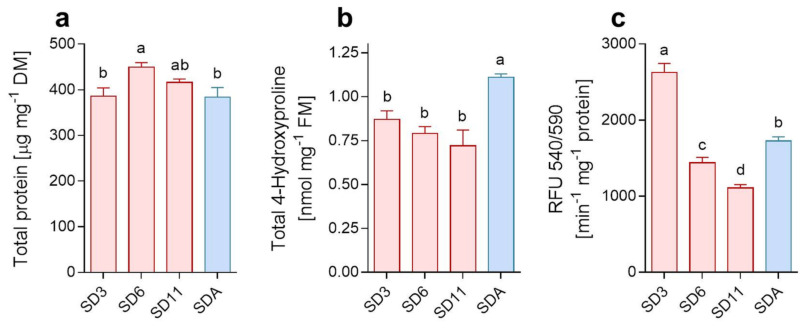
Total protein, collagens and MMPs. Changes in (**a**) total protein content, (**b**) total content of collagens estimated based on 4-hydroxyproline released upon acidic hydrolysis of larval total proteins, and (**c**) activity of general metalloproteinases (MMPs, collagenases). All analyses were performed on extracts of whole body larvae of different variants (diapause maintenance: SD3, SD6, SD1 (red columns); cold acclimation: SDA (blue column); see Table 1)) and each column is a mean of 3 biological replicates. Means were compared using one-way ANOVAs followed by Bonnferoni’s multiple comparison tests (different letters indicate significantly different means). Note that nanomolar amounts of hydroxyproline were released from collagens upon acidic hydrolysis (**c**), while the total pool of free hydroxyproline in larval tissues is three orders lower—approximately 4–8 femtomoles [22].

**Figure 3 metabolites-12-00163-f003:**
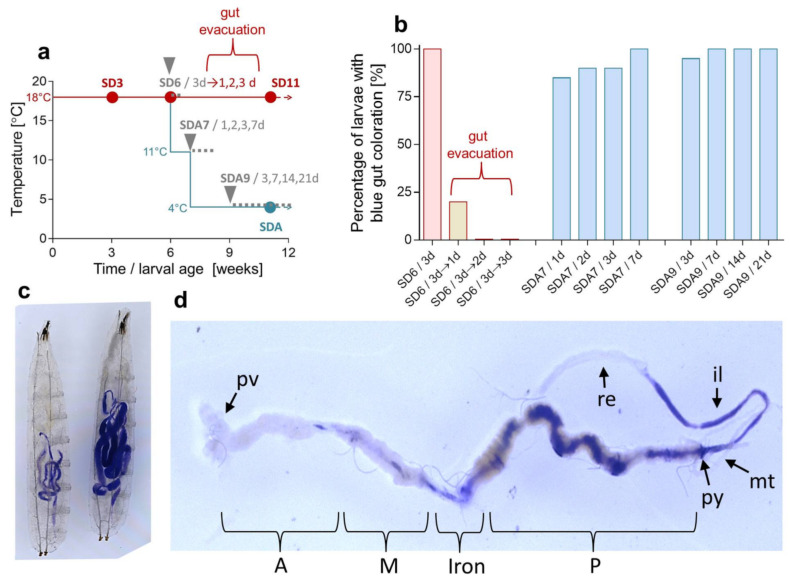
Diet consumption in diapause, of cold-acclimated larvae of *C. costata*. (**a**) shows the times when larvae were transferred to BB-augmented diets (grey triangles) and for how many days (grey font). Each column in (**b**) shows the percentage of larvae (*n* = 20 each) with blue gut contents. SD6 larvae exposed to BB diet for 3 d (all guts blue-colored) were returned to a standard diet to show that the blue coloration disappears in 80% of individuals within 1 d, and in all individuals within 2 d (gut evacuation). Most of the diapausing, cold-acclimated larvae (SDA7 and SDA9) exhibit blue guts within a few days of exposure to the BB diet. (**c**) shows examples of BB diet-fed larvae: left, an SDA9 larva exposed to BB diet for 14 d at 4 °C; right, a non-diapause (LD) larva exposed to BB diet for 1 h at 18 °C. (**d**) shows a gut dissected from an SDA larva exposed to the BB diet. Gut segments are described according to the gut morphology of *D. melanogaster* larvae: pv, proventriculus; A, M, P, anterior, middle, and posterior midgut; Iron, iron cells segment; py, pylorus; mt, Malpighian tubules; il, ileum; re, rectum.

**Figure 4 metabolites-12-00163-f004:**
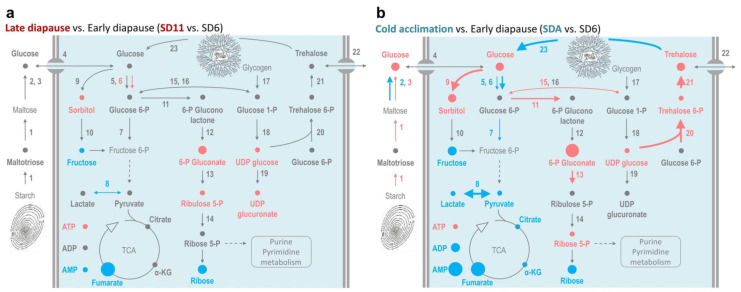
Changes in metabolism of trehalose during diapause maintenance vs. cold acclimation in *C. costata* larvae. The maps show log_2_-fold changes in relative concentrations of metabolites (circles) and gene expressions (arrows) during diapause maintenance (**a**) and gradual cold acclimation (**b**) (see Appendix A). Other descriptions are as in Figure 2. TCA, tricarboxylic acid cycle; α-KG, ketoglutarate.

**Figure 5 metabolites-12-00163-f005:**
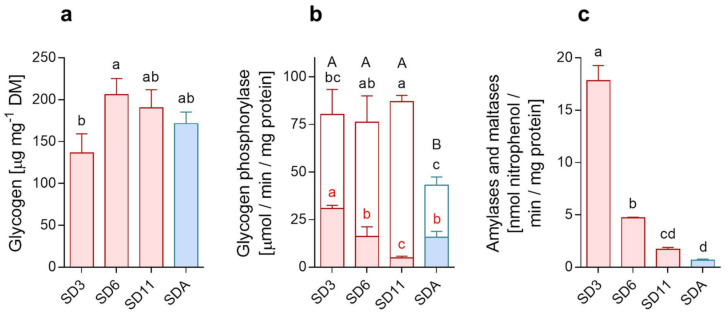
Glycogen, glycogen phosphorylase, amylases and maltases. Changes in (**a**) the total glycogen content, and the activities of (**b**) glycogen phosphorylase (white portions of the columns represent active form ‘a’ and colored portions inactive form ‘b’), and (**c**) amylases and maltases. All analyses were performed in extracts of whole body larvae of different acclimation variants (diapause maintenance: SD3, SD6, SD1 (red columns); cold acclimation: SDA (blue column); see Table 1)). Each column is a mean of 3 biological replicates. Means were compared using one-way ANOVAs followed by Bonnferoni’s multiple comparison tests (different letters indicate significantly different means; in panel b: the capital letters show differences between total activities, while the lower-case letters show differences between activities of the inactive (red font) and active (black font) forms of the glycogen phosphorylase).

**Figure 6 metabolites-12-00163-f006:**
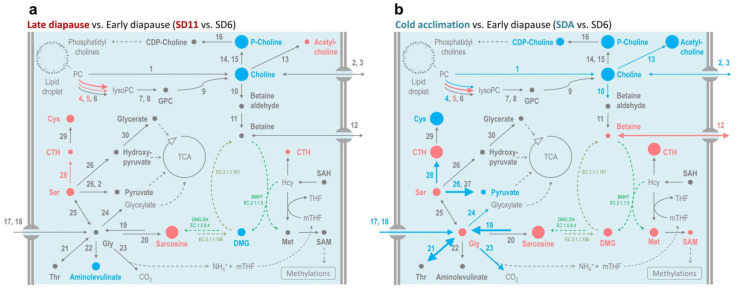
Changes in metabolism of betaine during diapause maintenance and cold acclimation. The maps show log2-fold changes in relative concentrations of metabolites (circles) and gene expressions (arrows) during diapause maintenance (**a**) and gradual cold acclimation (**b**) (see Appendix A). Other descriptions are as in Figure 2. TCA, tricarboxylic acid cycle; DMG, dimethylglycine; SAM, S-adenosylmethionine; SAH, S-adenosylhomocysteine; (m)THF, (methyl-) tetrahydrofolate; (lyso)PC, (lyso) phosphatidylcholine; GPC, glycerol 3-phosphocholine; BHMT (EC 2.1.1.5), betaine homocysteine methyltransferase; DMGDH (EC 1.5.8.4), DMG dehydrogenase; EC 2.1.1.156, sarcosine DMG methyltransferase; EC 2.1.1.161, DMG betaine methyltransferase.

**Figure 7 metabolites-12-00163-f007:**
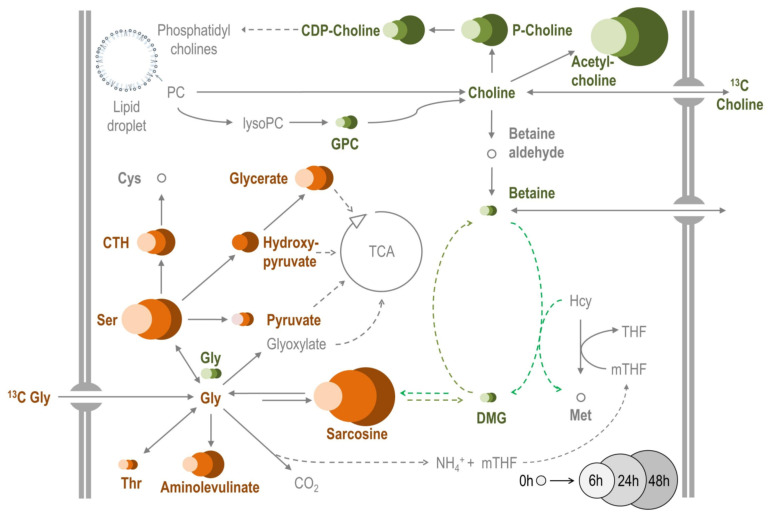
Fluxomics of potential metabolic precursors of betain in *C. costata* larvae. The map shows the metabolic destiny of two ^13^C-labelled compounds (^13^C-glycine, brown circles; 1,2-^13^C2 choline chloride, green circles) added to the diet of 15 d-old non-diapause (LD) larvae. The larvae were fed on these diets for 6 h, 24 h, or 48 h (6 h is the smallest and lightest circle on the left of the triplet) after which they were processed and analyzed using LC-HRMS metabolomic platform (for more detail, see Materials and Methods). The results are expressed for each metabolite as a percentage of the peak area of ^13^C-form (subtracting the native occurrence of ^13^C-carbon) relative to ^12^C-form (100% is the 0 h small circle in the legend). The size of circles is proportional to the percentage of the ^13^C-form. For reference to enzymes (arrows), see Figure 6.

**Figure 8 metabolites-12-00163-f008:**
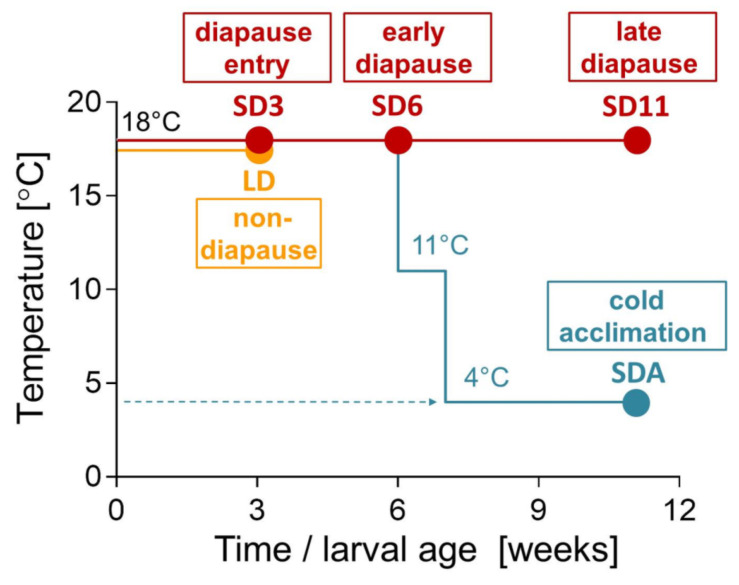
Phenotypic/acclimation variants of *Chymomyza costata*. *C. costata* flies were reared from eggs (time 0) to 3rd larval instars (time 3 weeks) at 18 °C under long-day photoperiod (LD; L16h:D8h, which allows direct non-diapause development to pupa and adult; orange line) or short-day photoperiod (SD; L12h:D12h, which induces diapause). Diapausing larvae then either remained at 18 °C and SD to maintain diapause (SD3, SD6, SD11; red line) or were transferred to constant darkness and progressively cold acclimated over five weeks (SDA; blue line). Note that non-diapause larvae (LD) were not used in this study except as controls for BB-food intake experiments and for ^13^C-precursor fluxomics. Filled circles indicate sample points for LC-HRMS metabolomics analyses and RNA extraction for RT-qPCR analysis of relative gene expression.

**Table 1 metabolites-12-00163-t001:** Freeze-tolerance phenotypes of *Chymomyza costata* larvae.

Phenotype	Acclimation Variant/ Phenotype	Freeze Tolerance at −30 °C * [% Larvae Surviving to Adulthood]	Survival in LN_2_ * [% Larvae Surviving to Adulthood]
SD3	Diapause entry	0	0
SD6	Early diapause maintenance	65.0	7.7
SD11	Late diapause maintenance	41.5	10.1
SDA	Diapause, cold-acclimated	75.9	42.5

* Data taken from [19,20,21].

## Data Availability

The data presented in this study are available in article and Appendix A.

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
