# Peer review of "Cryoprotective Metabolites Are Sourced from Both External Diet and Internal Macromolecular Reserves during Metabolic Reprogramming for Freeze Tolerance in Drosophilid Fly, Chymomyza costata"

_metabolites, 2022, doi:10.3390/metabo12020163_

Round 1

Reviewer 1 Report

The authors identify accumulation of cryoprotectants during development of freeze tolerance in larvae of Chymomyza costata.  The experiments are appropriate and the interpretation in terms of dietary and or metabolic sources sound.

My only suggestion is that interest for the general reader would be increased if some comments could be made about the possible involvement of different tissues in the metabolism.  I realise that the study of individual tissues would be impractical, but if there are any indications in existing literature, this could be mentioned in the Introduction.  Presumably all tissues need to accumulate cryoprotectants, but all might not be involved in their metabolism.  Perhaps in the Conclusion it might be possible to indicate which tissues would have made the largest contributions to the measured metabolites and enzymes

Author Response

We thank to Reviewer 1 for positive evaluation.

We agree with the reviewer that resolving the roles of different tissues, and their interactions, in biosynthesis and transport of CPs would be the ideal situation. Our metabolomic analyses, however, were conducted at the level of whole organism. Though seriously trying, we found that adding any information on tissue level into this particular paper is problematic: either too speculative if we speak only about specific case of C. costata, or too extensive if we want to speak more broadly about insects (or even organisms).

In order to at least somehow reflect the (very correct) suggestion of the reviewer, we added a short explanation into the first paragraph of Results and Discusssion section: 

"It is important to note that our schematic maps do not represent any specific cell type though they show plasmatic membranes and distinguish between cytosolic and extracellular spaces. As the metabolomic analyses were conducted using whole larvae, resolution to tissue level is not possible."

Reviewer 2 Report

This is a comprehensive study that investigates the metabolic pathways for insect cold tolerance, and could also serve as a reference to similar mechanisms in other taxonomy such as plants or bacteria. This research benefits from the previous works from the same group in which different freeze-tolerance phenotypes of C. costata were generated. In the present study, the contents of the cryoprotectant metabolites and the expression of genes in the responsive pathways were measured and compared among those freeze-tolerance phenotypes.  One of the key cryoprotective metabolite, proline, was found to be accumulated in the freeze-tolerance phenotype which were sourced from both glutamine reserves and ingested food. The diet consumption experiments and LC-HRMS analysis of 13C-labelled compounds were also well carried out to support their conclusions. This is a solid and well-presented study. I have a few comments as below:

Title: the question style (should followed by a question mark then) not really captures the essence of this study, importantly, the species that was investigated. I would suggest “Cryoprotective metabolites are sourced from both internal macromolecular reserves and external compound-enriched diet for freezing-tolerance in the Drosophilid fly, Chymomyza costata” or something similar for the authors to consider.

Abstract line 26: “Here we resolve the metabolism…” is a rather bold claim. Better “Here we aim to resolve the metabolism…”. Also the abstract gives too much results but misses the discussion and conclusion.

Introduction: the authors spent a lot of time to describe the cold metabolism in animals, plants, bacteria and insect in general, but it is not elaborated enough why C. costata was chosen in this study. The authors briefly cited their previous reports but this story should be able to stand by itself. Thereby, more background of the previous finding should be given to sufficiently bring the audience up to speed why C. costata is an ideal model here.

Line 114: add “previously” before “generated” to emphasize that this is from published studies, despite showing it in the “Results” section.

Line 67, 83, 160, 176, 185, 244, 258, 285 and so on: check the excessive space between words.  

Line 286: change “consistenr” to “consistent”.

Fig.2,Fig.3b,Fig.5: why SDA bar is in blue and others in red? Didn’t explain in the figure or legend.

Fig.5: the statistic letters have to be specifically defined for each variable. Now the lower-case letters are used for two sets of data which should be avoided.

Author Response

We thank to reviewer 2 for positive evaluation and, particularly, for very helpful comments.

We accepted practically all suggestions made by reviewer 2 and changed our manuscript accordingly. The point-to-point response is in the attached file.

Thanks again as we feel that the changes we made improve the quality of our paper.
